# Investigating Human Priors for Playing Video Games

**Rachit Dubey, Pulkit Agrawal, Deepak Pathak, Thomas L. Griffiths, and Alexei A. Efros**

University of California, Berkeley

## Abstract

What makes humans so good at solving seemingly complex video games? Unlike computers, humans bring in a great deal of prior knowledge about the world, enabling efficient decision making. This paper investigates the role of human priors for solving video games. Given a sample game, we conduct a series of ablation studies to quantify the importance of various priors. We do this by modifying the video game environment to systematically mask different types of visual information that could be used by humans as priors. We find that removal of some prior knowledge causes a drastic degradation in the speed with which human players solve the game, e.g. from 2 minutes to over 20 minutes. Furthermore, our results indicate that general priors, such as the importance of objects and visual consistency, are critical for efficient game-play.

## 1 Introduction

While deep Reinforcement Learning (RL) methods have shown impressive performance on a variety of video games (Mnih et al., 2015), they remain woefully inefficient compared to human players, taking millions of action inputs to solve even the simplest Atari games. Much research is currently focused on improving sample efficiency of RL algorithms (Oh et al., 2017; Gu et al., 2016). However, there is an orthogonal issue that is often overlooked: RL agents attack each problem *tabula rasa*, whereas humans come in with a wealth of prior knowledge about the world, from physics to semantics to affordances.

Consider the following motivating example: you are tasked with playing an unfamiliar computer game shown in Figure 1(a). No manual or instructions are provided; you don't even know which game sprite is controlled by you. Indeed, the only feedback you are ever given is "terminal", i.e. once you successfully finish the game. Would you be able to successfully finish this game? How long would it take? We recruited forty human subjects to play this game and found that subjects finished it quite easily, taking just under 1 minute of game-play or  3000 action inputs. This is not overly surprising as one could easily guess that the game's goal is to move the robot sprite towards the princess by stepping on the brick-like objects and using ladders to reach the higher platforms while avoiding the angry pink and the fire objects.

Now consider a second scenario in which this same simple game is re-rendered with new textures, getting rid of semantic and affordance (Gibson, 2014) cues, as shown in Figure 1(b). How would human performance change? We recruited another forty subjects to play this game and found that, on average, it took the players more than twice the time (2 minutes) and action inputs ( 6500) to complete the game. The second game is clearly much harder for humans, likely because it is now more difficult to guess the game structure and goal, as well as to spot obstacles.

For comparison, we can also examine how modern RL algorithms perform on these games. This is not so simple, as most standard RL approaches expect very dense rewards (e.g. continuously updated game-score (Mnih et al., 2015)), whereas we provide only a terminal reward, to mimic how most humans play video games. In such sparse reward scenarios, standard methods like A3C (Mnih et al., 2016) are too sample-inefficient and were too slow to finish the games. Hence, we used a curiosity-based RL algorithm specifically tailored to sparse-reward settings (Pathak et al., 2017), which was able to solve both games. Unlike humans, RL did not show much difference between the

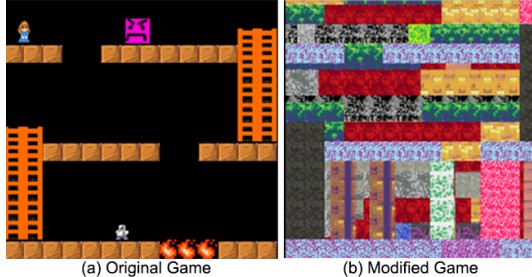
(a) Original Game   (b) Modified Game

Figure 1: **Motivating example.** (a) A simple platformer game. (b) The same game modified by re-rendering the textures. Despite the two games being structurally the same, human players took twice as long to finish the second game as the first one. In comparison, the performance of an RL agent was approximately the same for the two games.

two games, taking about 4 million action inputs to solve each one. This should not be surprising: since RL did not have any prior knowledge about the world, both these games carried roughly the same amount of information from the perspective of the agent.

This simple motivating experiment highlights the importance of prior knowledge that humans draw upon to quickly solve tasks given to them (Lake et al., 2016; Tsividis et al., 2017). Developmental psychologists have begun documenting the prior knowledge that children draw upon in learning about the world (Spelke & Kinzler, 2007; Carey, 2009). However, these studies have not explicitly quantified the relative importance of the various priors for problem-solving.

In this work, we systematically quantify the importance of different types of priors humans bring to bear while solving one particular kind of problem – video games. We chose video games as the task for our investigation because it is relatively easy to methodically change the game to include or mask different kinds of knowledge and run large-scale human studies. Furthermore, video games, such as ATARI, are a popular choice in the reinforcement learning community.

The paper consists of a series of ablation studies on a specially-designed game environment, systematically masking out various types of visual information that could be used by humans as priors. The full game (unlike the motivating example above) was designed to be sufficiently complex and difficult for humans to easily measure changes in performance between different testing conditions.

We find that removal of some prior knowledge causes a drastic degradation in the performance of human players from 1 minute to over 20 minutes. Another key finding of our investigation is that while specific knowledge, such as "ladders are to be climbed", "keys are used to open doors", "jumping on spikes is dangerous", is important for humans to quickly solve games, more general priors about the importance of objects and visual consistency are even more critical.

## 2  METHOD

To investigate the aspects of visual information that enable humans to efficiently solve video games, we designed a browser-based platform game consisting of an agent sprite, platforms, ladders, angry pink object that kills the agent, spikes that are dangerous to jump on, a key, and a door (see Figure 2 (a)). The agent sprite can be moved with the help of arrow keys. A terminal reward of +1 is provided when the agent reaches the door after having to taken the key, thereby terminating the game. The game is reset whenever the agent touches the enemy, jumps on the spike, or falls below the lowest platform. We made this game to resemble the exploration problems faced in the classic ATARI game of *Montezuma's Revenge* that has proven to be very challenging for deep reinforcement learning techniques (Bellemare et al., 2016; Mnih et al., 2015). Unlike the motivating example, this game is too large-scale to be solved by RL agents, but provides the complexity we need to run a wide range of human experiments.

We created different versions of the video game by re-rendering various entities such as ladders, enemies, keys, platforms etc. using alternate textures (Figure 2). These textures were chosen to mask various forms of prior knowledge that are described in the experiments section. We also changed various physical properties of the game, such as the effect of gravity, and the way the agent interacts with its environment. Note that all the games were exactly the same in their underlying

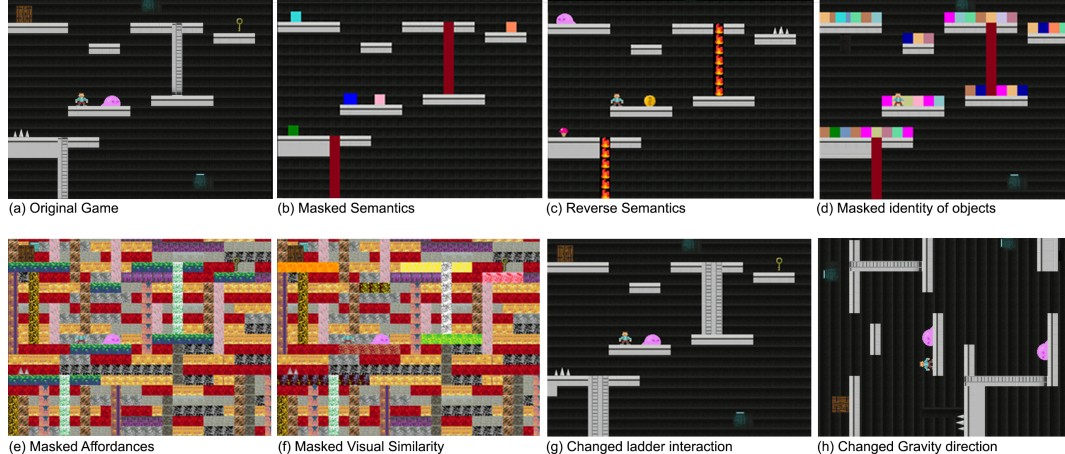

(a) Original Game     (b) Masked Semantics     (c) Reverse Semantics     (d) Masked identity of objects

(e) Masked Affordances     (f) Masked Visual Similarity     (g) Changed ladder interaction     (h) Changed Gravity direction

Figure 2: **Various game manipulations**. (a) Original version of the game. (b) Game with masked objects to ablate semantics prior. (c) Game with reversed associations as an alternate way to ablate semantics prior. (d) Game with masked objects and distractor objects to ablate the concept of object. (e) Game with background textures to ablate affordance prior. (f) Game with background textures and different colors for all platforms to ablate similarity prior. (g) Game with modified ladder to hinder participant's prior about ladder interactions. (h) Rotated game to change participant's prior about gravity.

structure and reward, as well as the shortest path to reach the goal, thereby ensuring that the change in human performance (if any) is only due to masking of the priors.

We quantified human performance on each version of the game by recruiting 120 participants from Amazon Mechanical Turk. Each participant was instructed to finish the game as quickly as possible using the arrow keys as controls, but *no information* about the goals or the reward structure of the game was communicated. Each participant was paid $1 for successfully completing the game. The maximum time allowed for playing the game was set to 30 minutes. For each participant, we recorded the $(x, y)$ position of the player at every step of the game, the total time taken by the participant to finish the game and the total number of deaths before finishing the game. We used this data to quantify the performance of each participant. Note that each participant was only allowed to complete a game once, and could not participate again (i.e. different 120 participants played each version of the game).

## 3   QUANTIFYING THE IMPORTANCE OF OBJECT PRIORS

The original game (available to play at this link) is shown in Figure 2(a). A single glance at this game is enough to inform human players that the agent sprite has to reach the key to open the door while avoiding the dangerous objects like spikes and angry pink slime. Unsurprisingly, humans quickly solve this game. Figure 3(a) shows that the average time taken to complete the game is 1.8 minutes (blue bar) and the average number of deaths (3.3, orange bar) and unique game states visited (3011, yellow bar) are all quite small.

### 3.1   SEMANTICS

To study the importance of prior knowledge about object semantics, we rendered objects and ladders with blocks of uniform color as shown in Figure 2(b). This game can be played at this link. In this version, the visual appearance of objects conveys no information about their semantics. Results in Figure 3(b) show that human players take more than twice the time (4.3 minutes), have higher number of deaths (11.1), and explore significantly larger number of states (7205) as compared to the original game (p-value: $p < 0.01$). This clearly demonstrates that masking semantics hurts human performance.

A natural question is how do humans make use of semantic information? One hypothesis is that knowledge of semantics enables humans to infer the latent reward structure of the game. If this

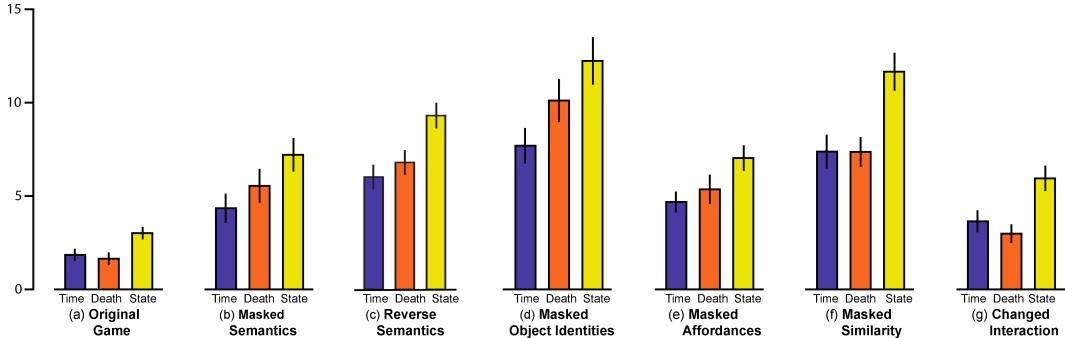

Figure 3: **Quantifying the influence of various object priors.** The blue bar shows average time taken by humans (in minutes), orange bar shows the average number of deaths, and yellow bar shows the number of unique states visited by players to solve the various games. For visualization purposes, the number of deaths is divided by 2, and the number of states is divided by 1000 respectively.

indeed is the case, then in the original game, where the key and the door are both visible, players should first visit the key and then go to the door, while in the version of the game without semantics, players should not exhibit such bias. We found that in the original game, nearly all participants reached the key first, while in the version with masked semantics only $42$ out of $120$ participants reached the key before the door (see Figure 4(a)). Moreover, human players took significantly longer to reach the door after taking the key as compared to the original game (see Figure 4(b)). This result provides further evidence that in the absence of semantics, humans are unable to infer the reward structure and consequently significantly increase their exploration. To rule out the possibility that increase in time is simply due to the fact players take longer to finish the game without semantics, the time to reach the door after taking the key was normalized by the total amount of time spent by the player to complete the game.

To further quantify the importance of semantics, instead of simply masking, we manipulated the semantic prior by swapping the semantics between different entities. As seen on Figure 4(c), we replaced the pink enemy and spikes by coins and ice-cream objects respectively which have a positive connotation; the ladder by fire, the key and the door by spikes and enemies which have negative connotations (see game link). As shown in Figure 3(c), the participants took longer to solve this game ($6.1$ minutes, $p < 0.01$). The average number of deaths ($13.7$) was also significantly more and the participants explored more states ($9400$) compared to the original version ($p < 0.01$ for both). Interestingly, the participants also took longer compared to the masked semantics version ($p < 0.05$) implying that when we reverse semantic information, humans find the game even tougher.

## 3.2 OBJECTS AS SUB-GOALS FOR EXPLORATION

While blocks of uniform color in the game shown in Figure 2(b) convey no semantics, they are distinct from the background and seem to attract human attention. It is possible that humans infer these distinct entities (or objects) as sub-goals, which results in more efficient exploration than random search. That is, there is something special about objects that draws human attention compared to any random piece of texture. To test this, we modified the game to cover each space on the platform with a block of different color to hide where the objects are (see Figure 2(d), game link). Most colored blocks are placebos and do not correspond to any object and the actual objects have the same color and form as in the previous version of the game with masked semantics (i.e., Figure 2(b)). If the prior knowledge that visibly distinct entities are interesting to explore is critical, this game manipulation should lead to a significant drop in human performance.

Results in Figure 3(d) show that masking the concept of objects leads to drastic deterioration in performance. The average time taken by human players to solve the game is nearly four times longer ($7.7$ minutes), the number of deaths is nearly six times greater ($20.2$), and humans explore four times as many game states ($12,232$) as compared to the original game. When compared to the game version in which only semantic information was removed (Figure 3(b)), the time taken, number of deaths and number of states are all significantly greater ($p < 0.01$). When only semantics are removed, after encountering one object, human players become aware of what possible locations

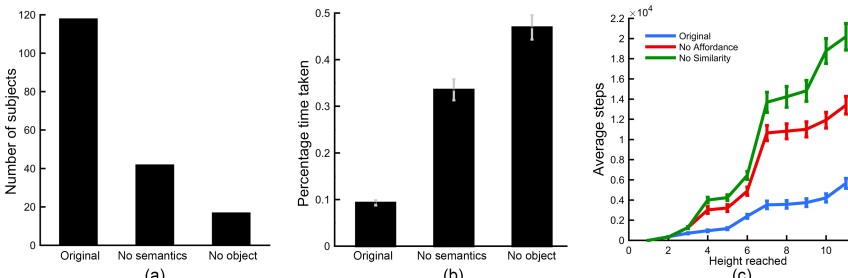

Figure 4: **Change in behavior upon ablation of various priors**. (a) Graph comparing number of participants that reached the key before the door in the original version, game without semantics, and game without object prior. (b) Amount of time taken by participants to reach the door once they obtained the key. (c) Average number of steps taken by participants to reach various vertical levels in original version, game without affordance, and game without similarity.

might be interesting to explore next. However, when concept of objects is also masked, it is unclear what to explore next. This effect can be seen by the increase in normalized time taken to reach the door from the key as compared to the game where only semantics are masked (Figure 4(b)). All these results suggest that concept of objects i.e. knowing that visibly distinct entities are interesting and can be used as sub-goals for exploration, is a critical prior and perhaps more important than knowledge of semantics.

### 3.3 AFFORDANCES

Until now, we manipulated objects in ways that made inferring the underlying reward structure of the game non-trivial. However, in these games it was obvious for humans that *platforms* can support agent sprites, *ladders* could be climbed to reach different platforms (even when the ladders were colored in uniform red in games shown in Figure 2(b,c), the connectivity pattern revealed where the ladders were) and black parts of the game constitute *free space*. Here, the platforms and ladders *afford* the actions of walking and climbing (Gibson, 2014), irrespective of their appearance. In the next set of experiments, we manipulated the game to mask the affordance prior.

One way to mask affordances is to fill free space with random textures, which are visually similar to textures used for rendering ladders and platforms (see Figure 2(e), game link). Note that in this game manipulation, objects and their semantics are clearly observable. When tasked to play this game, as shown in Figure 3(e), humans require significantly more time (4.7 minutes), die more often (10.7), and visit more states (7031) compared to the original game ($p < 0.01$). On the other hand, there is no significant difference in performance compared to the game without semantics, i.e., Figure 2(b), implying that the affordance prior is as important as the semantics prior in our setup.

### 3.4 THINGS THAT LOOK SIMILARLY, BEHAVE SIMILARLY

In the previous game, although we masked affordance information, once the player realizes that it is possible to stand on a particular texture and climb a specific texture, it is easy to use color/texture similarity to identify other platforms and ladders in the game. Similarly, in the game with masked semantics (Figure 2(b)), visual similarity can be used to identify other enemies and spikes. These considerations suggest that a general prior of the form that things that look the same act the same might help humans efficiently explore environments where semantics or affordances are hidden.

We tested this hypothesis by modifying the masked affordance game in a way that none of the platforms and ladders had the same visual signature (Figure 2(f), game link). Such rendering prevented human players from using the similarity prior. Figure 3(f)) shows that performance of humans was significantly worse in comparison to the original game (Figure 2(a)), the game with masked semantics (Figure 2(b)) and the game with masked affordances (Figure 2(e)) ($p < 0.01$). When compared to the game with no object information (Figure 2(d)), the time to complete the game (7.6 minutes) and the number of states explored by players were similar ($11, 715$), but the number of deaths ($14.8$) was significantly lower ($p < 0.01$). These results suggest that visual similarity is the second most important prior used by humans in gameplay after the knowledge of directing exploration towards objects.

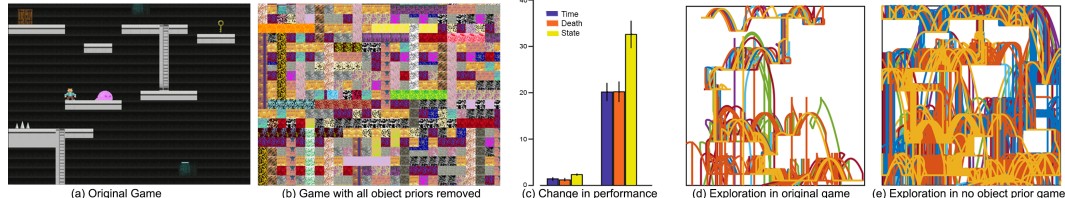

(a) Original Game  (b) Game with all object priors removed  (c) Change in performance  (d) Exploration in original game  (e) Exploration in no object prior game

Figure 5: **Masking all object priors drastically affects human performance**. (a) Original game. (b) Version without any object priors. (c) Graph depicting difference in participant's performance for both the games. (d) Exploration trajectory for original version and (e) for no object prior version.

In order to gain insight into how this prior knowledge affects humans, we investigated the exploration pattern of human players. In the game when all information is visible we expected that the progress of humans would be uniform in time. In the case when affordances are removed, the human players would initially take some time to figure out what visual pattern corresponds to what entity and then quickly make progress in the game. Finally, in the case when the similarity prior is removed, we would expect human players to be unable to generalize any knowledge across the game and to take large amounts of time exploring the environment even towards the end. We investigated if this indeed was true by computing the time taken by each player to reach different vertical distances in the game for the first time. Note that the door is on the top of the game, so the moving up corresponds to getting closer to solving the game. The results of this analysis are shown in Figure 4(c). The horizontal-axis shows the height reached by the player and the vertical-axis show the average time taken by the players. As the figure shows, the results confirm our hypothesis.

### 3.5 How to interact with objects

Until now we have analyzed the prior knowledge used by humans to interpret the visual structure in the game. However, interpretation of visual structure is only useful if the player understands what to do with the interpretation. Humans seem to possess prior knowledge about how to interact with different objects. For example, monsters can be avoided by jumping over them, ladders can be climbed by pressing the up key repeatedly etc. Deep reinforcement learning agents, on the other hand, do not possess such priors and must learn how to interact with objects by mere trial and error.

To test how critical such prior knowledge is, we created a version of the game in which the ladders couldn't be climbed by simply pressing the up key. Instead, the ladders were zigzag in nature and in order to climb the ladder players had to press the up key, followed by alternating presses between the right and left key. Note that the ladders in this version looked like normal ladders, so players couldn't infer the properties of the ladder by simply looking at them (see Figure 2(g), game link). As shown in Figure 3(g), changing the property of the ladder increases the time taken (3.6 minutes), number of deaths (6), and states explored (5942) when compared to the original game ($p < 0.01$).

## 4 Taxonomy of object priors

In previous sections, we studied how different priors about objects affect human performance one at a time. To quantify human performance when all object priors investigated so far are simultaneously masked, we created the game shown in Figure 5(b) that hid all information about objects, semantics, affordance, and similarity(game link). Results in Figure 5(c) show that humans found it extremely hard to play this game. The average time taken to solve the game increased to 20 minutes and the average number of deaths rose sharply to 40. Remarkably, the exploration trajectory of humans is now almost completely random as shown in Figure 5(e) with the number of unique states visited by the human players increasing by a factor of 9 as compared to the original game. Due to difficulty in completing this game, we noticed a high dropout of human participants before they finished the game. We had to increase the pay to $2.25 to encourage participants not to quit. Many participants noted that they could solve the game only by memorizing it.

Even though we preserved priors related to physics (e.g., objects fall down) and motor control (e.g., pressing left key moves the agent sprite to the left), just by rendering the game in a way that makes it impossible to use prior knowledge about how to visually interpret the game screen makes the game extremely hard to play. To further test the limits of human ability, we designed a harder

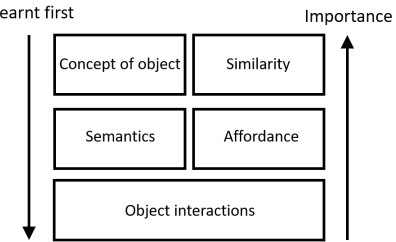

Figure 6: **Taxonomy of object priors.** The earlier an object prior is obtained during childhood, the more critical that object prior is in human problem solving in video games.

game where we also reversed gravity and randomly re-mapped the key presses to how it affect's the motion of agent's sprite. We, the creators of the game, having played a previous version of the game hundreds of times had an extremely hard time trying to complete this version of the game. This game placed us in the shoes of reinforcement learning (RL) agents that start off without the immense prior knowledge that humans possess. While improvements in the performance of RL agents with better algorithms and better computational resources is inevitable, our results make a strong case for developing algorithms that incorporate prior knowledge as a way to improve the performance of artificial agents.

While there are many possible directions on how to incorporate priors in RL and more generally AI agents, it is informative to study how humans acquire such priors. Studies in developmental psychology suggest that human infants as young as 2 months old possess a primitive notion of objects and expect them to move as connected and bounded wholes that allows them to perceive object boundaries and therefore possibly distinguish them from the background (Spelke, 1990; Spelke & Kinzler, 2007). At this stage, infants do not reason about object categories. By the age of 3-5 months, infants start exhibiting categorization behavior based on similarity and familiarity (Mandler, 1998; Mareschal & Quinn, 2001). The ability to recognize individual objects rapidly and accurately emerges comparatively late in development (usually by the time babies are 18-24 months old (Pereira & Smith, 2009)). Similarly, while young infants exhibit some knowledge about affordances early during development, the ability to distinguish a walkable step from a cliff emerges only by the time they are 18 months old (Kretch & Adolph, 2013).

These results in infant development suggest that starting with a primitive notion of objects, infants gradually learn about visual similarity and eventually about object semantics and affordances. It is quite interesting to note that the order in which infants increase their knowledge matches the importance of different object priors such as the existence of objects as sub-goals for exploration, visual similarity, object semantics, and affordances. Based on these results, we suggest a possible taxonomy and ranking of object priors in Figure 6. We put 'object interaction' at the bottom as in the context of our problem, knowledge about how to interact with specific objects can be only learned once recognition is performed.

## 5 PHYSICS AND MOTOR CONTROL PRIORS

In addition to prior knowledge about objects, humans also bring in rich prior knowledge about intuitive physics and strong motor control priors when they approach a new task (Hespos et al., 2009; Baillargeon, 2004; 1994; Wolpert & Ghahramani, 2000). Here, we have taken some initial steps to explore the importance of such priors in context of human gameplay.

### 5.1 GRAVITY

One of the most obvious forms of knowledge that we have about the physical world is with regards to gravity, i.e., things fall from up to down. To mask this prior, we created a version of the game in which the whole game window was rotated 90° (refer to Figure 2(h)). In this way, the gravity was reversed from left to right (as opposed to up to down). As shown in Figure 7, participants spent more time to solve this game compared to the original version with average time taken close to 3 minutes ($p < 0.01$). The average number of deaths and number of states explored was also significantly larger than the original version ($p < 0.01$).

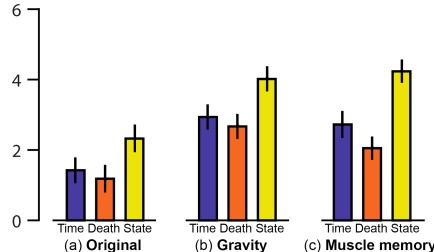

Figure 7: **Quantifying physics and motor control priors.** Graph shows performance of participants in original version, game with gravity reversed, and game with key controls reversed. Number of deaths is divided by 2 and number of states is divided by 1000.

## 5.2 MUSCLE MEMORY

Human players also come with knowledge about the consequences of actions such as pressing arrow keys moves the agent sprite in the corresponding directions (i.e., pressing up makes the agent sprite jump, pressing left makes the agent sprite go left and so forth). We created a version of the game in which we reversed the arrow key controls. Thus, pressing the left arrow key made the agent sprite go right, pressing the right key moved the sprite left, pressing the down key made the player jump (or go up the stairs), and pressing the up key made the player go down the stairs. Participants again took longer to solve this game compared to the original version with average time taken close to 3 minutes (refer to Figure 7). The average number of deaths and number of states explored was also significantly larger than the original version ($p < 0.01$). Interestingly, the performance of players when the gravity was reversed, and key controls were reversed is similar, with no significant difference between the two conditions.

## 6 CONTROLLING FOR CHANGE IN COMPLEXITY

So far in this paper, we have manipulated various visual priors while keeping the underlying game and reward structure exactly the same. We have assumed that this will influence human performance while keeping RL agent performance unchanged, since RL does not have any priors to begin with. However, one possible confound is that the visual complexity of the modified games might have changed from the original game version, because masking out priors without changing visual complexity is extremely difficult.

To control for this confound, we investigated the performance of an RL agent on the various game manipulations. If RL agents are not affected by the game manipulations, then it would suggest that prior knowledge and not visual complexity is the main reason behind the change in human performance. Note that this confound is not present in the physics and motor control experiments as the visual input stays the same as the original game.

To this end, we systematically created different versions of the game in Figure 1(a) to ablate semantics, the concept of object, affordance, and similarity as shown in Figure 8. Note that the game used for human experiments shown in Figure 2 is more complex than the game used for RL experiments in Figure 8. This is because the larger game was simply too hard for state-of-the-art RL agents to solve. Apart from the difference in the game size, we tried to make the games as similar as possible. Even though this version of the game is simpler (regarding size, number of objects etc.), we note that this game is still non-trivial for an RL agent. For instance, due to the sparse reward structure of the game, both A3C (Mnih et al., 2016) and breadth-first search didn't come close to solving the game even after 10 million steps. Hence, for our purpose, we used an RL algorithm augmented with a curiosity based exploration strategy (Pathak et al., 2017). For each game version, we report the mean performance of five random seeds that succeeded.

As shown in Figure 8(e), the RL agent was unaffected by the removal of semantics, the concept of objects, as well as affordances – there is no significant difference between the mean score of the RL agent on these games when compared to the performance on the original game ($p > 0.05$). This suggests that the drop in human performance in these game manipulations is not due to the change in visual complexity, but it is rather due to the masking of the various priors. On the other hand, the performance of the RL agent does worsen when visual similarity is masked as it takes nearly twice as many interactions to complete the game compared to the original version. We believe this is due

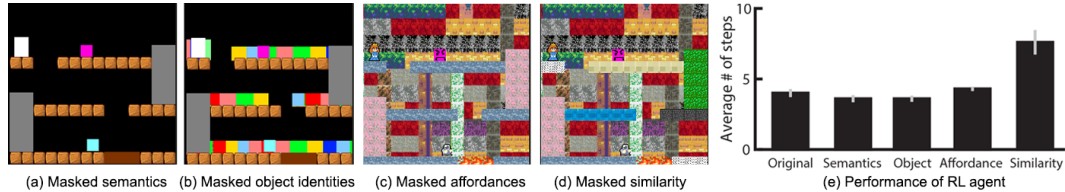

(a) Masked semantics    (b) Masked object identities    (c) Masked affordances    (d) Masked similarity    (e) Performance of RL agent

Figure 8: **Quantifying the performance of RL agent.** (a) Game without semantic information. (b) Game with masked and distractor objects to ablate concept of objects. (c) Game without affordance information. (d) Game without similarity information. (e) Performance of RL agent on various game manipulations (steps shown in order of million). Error bars indicate standard error of mean for the 5 random seeds. The RL agent performs similarly on all games except for the one without visual similarity.

to to the use of *convolutional* neural networks that implicitly impose the prior of visual similarity rather than simply due to the change in visual complexity.

## 7 DISCUSSION

While there is no doubt that the performance of deep RL algorithms is impressive, there is much to be learned from human cognition if our goal is to enable RL agents to solve sparse reward tasks with human-like efficiency. Humans have the amazing ability to use their past knowledge (i.e., priors) to solve new tasks quickly. Success in such scenarios critically depends on the agent's ability to explore its environment and then promptly learn from its successes (Daw et al., 2006; Cohen et al., 2007). In this vein, our results demonstrate the importance of prior knowledge in helping humans explore efficiently in these sparse reward environments (Knox et al., 2012; Gershman & Niv, 2015).

However, being equipped with strong prior knowledge can sometimes lead to constrained exploration that might not be optimal in all environments (Lucas et al., 2014; Bonawitz et al., 2011). For instance, consider the game shown in Figure 9 consisting of a robot and a princess object. The game environment also includes rewards in hidden locations (shown as dashed yellow boxes only for illustration). When tasked to play this game, human participants (n=30) immediately assume that princess is the goal and do not explore the free space containing hidden rewards. They directly reach the princess and thereby terminate the game with sub-optimal rewards. In contrast, a random agent (30 seeds) ends up obtaining almost four times more reward than human players as shown in Figure 9. Thus, while incorporating prior knowledge in RL agents has many potential benefits, future work should also consider challenges regarding under-constrained exploration in certain kinds of settings.

While our paper primarily investigated object priors (and physics priors to some extent), humans also possess rich prior knowledge about the world in the form of intuitive psychology and also bring in various priors about general video game playing such as that moving up and to the right in games is generally correlated with progress, games have goals, etc. Studying the importance of such priors will be an interesting future direction of research.

Building RL algorithms that require fewer interactions to reach the goal (i.e., sample efficient algorithms) is an active area of research, and further progress is inevitable. In addition to developing better optimization methods, we believe that instead of always initializing learning from scratch, either incorporating prior knowledge directly or constructing mechanisms for condensing experience into reusable knowledge (i.e., learning priors through *continual* learning) might be critical for building RL agents with human-like efficiency. Our work takes first steps toward quantifying the

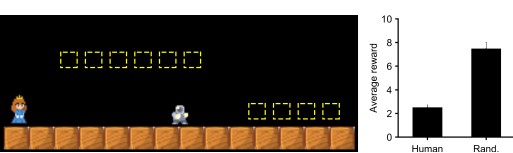

Figure 9: **Prior information constrains human exploration.** (Left) A very simple game with hidden rewards (shown in dashed yellow). (Right) Average rewards accumulated by human players vs a random agent.

importance of various priors that humans employ in solving video games and in understanding how prior knowledge makes humans good at such complex tasks. We believe that our results will inspire researchers to think about different mechanisms of incorporating prior knowledge in the design of RL agents. We also hope that our experimental platform of video games, available in open-source, will fuel more detailed studies investigating human priors and a benchmark for quantifying the efficacy of different mechanisms of incorporating prior knowledge into RL agents.

## ACKNOWLEDGEMENT

We thank Jordan Suchow, Michael Chang, Shubham Tulsiani, Alison Gopnik, and other members of the BAIR community for helpful discussions and comments. This work has been supported, in part, by Google, ONR MURI N00014-14-1-0671, Berkeley DeepDrive, NVIDIA Graduate Fellowship to DP, and the Valrhona Reinforcement Learning Fellowship.

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
