# OpenReview forum: "Investigating Human Priors for Playing Video Games"
_ICLR.cc/2018/Conference — Invite to Workshop Track_

### Official Review · AnonReviewer1 · 2017-11-21
**Investigating Human Priors review**

**Rating:** 5
**Confidence:** 4

**Review:**

Overall:
I really enjoyed reading this paper and think the question is super important. I have some reservations about the execution of the experiments as well as some of the conclusions drawn. For this reason I am currently a weak reject (weak because I believe the question is very interesting). However, I believe that many of my criticisms can be assuaged during the rebuttal period.

Paper Summary:
For RL to play video games, it has to play many many many many times. In fact, many more times than a human where prior knowledge lets us learn quite fast in new (but related) environments. The authors study, using experiments, what aspects of human priors are the important parts.

The authors’ Main Claim appears to be: “While common wisdom might suggest that prior knowledge about game semantics such as ladders are to be climbed, jumping on spikes is dangerous or the agent must fetch the key before reaching the door are crucial to human performance, we find that instead more general and high-level priors such as the world is composed of objects, object like entities are used as subgoals for exploration, and things that look the same, act the same are more critical.”

Overall, I find this interesting. However, I am not completely convinced by some of the experimental demonstrations.

Issue 0: The experiments seem underpowered / not that well analyzed.
There are only 30 participants per condition and so it’s hard to tell whether the large differences in conditions are due to noise and what a stable ranking of conditions actually looks like. I would recommend that the authors triple the sample size and be more clear about reporting the outcomes in each of the conditions.

It’s not clear what the error bars in figure 1 represent, are they standard deviations of the mean? Are they standard deviations of the data? Are they confidence intervals for the mean effect?

Did you collect any extra data about participants? One potentially helpful example is asking how familiar participants are with platformer video games. This would give at least some proxy to study the importance of priors about “how video games are generally constructed” rather than priors like “objects are special”.

Issue 1: What do you mean by “objects”?
The authors interpret the fact that performance falls so much between conditions b and c to mean that human priors about “objects are special” are very important. However, an alternative explanation is that people explore things which look “different” (ie. Orange when everything else is black).

The problem here comes from an unclear definition of what the authors mean by an “object” so in revision I would like authors to clarify what precisely they mean by a prior about “the world is composed of objects” and how this particular experiment differentiates “object” from a more general prior about “video games have clearly defined goals, there are 4 clearly defined boxes here, let me try touching them.”

This is important because a clear definition will give us an idea for how to actually build this prior into AI systems.

Issue 2: Are the results here really about “high level” priors?
There are two ways to interpret the authors’ main claim: the strong version would maintain that semantic priors aren’t important at all.

There is no real evidence here for the strong version of the claim. A real test would be to reverse some of the expected game semantics and see if people perform just as well as in the “masked semantics” condition.

For example, suppose we had exactly the same game and N different types of objects in various places of the game where N-1 of them caused death but 1 of them opened the door (but it wasn’t the object that looked like a key). My hypothesis would be that performance would fall drastically as semantic priors would quickly lead people in that direction.

Thus, we could consider a weaker version of the claim: semantic priors are important but even in the absence of explicit semantic cues (note, this is different from having the wrong semantic cues as above) people can do a good job on the game. This is much more supported by the data, but still I think very particular to this situation. Imagine a slight twist on the game:

There is a sword (with a lock on it), a key, a slime and the door (and maybe some spikes). The player must do things in exactly this order: first the player must get the key, then they must touch the sword, then they must kill the slime, then they go to the door. Here without semantic priors I would hypothesize that human performance would fall quite far (whereas with semantics people would be able to figure it out quite well).

Thus, I think the authors’ claim needs to be qualified quite a bit. It’s also important to take into account how much work general priors about video game playing (games have goals, up jumps, there is basic physics) are doing here (the authors do this when they discuss versions of the game with different physics).

---

> ### Author Response · Authors · 2018-01-05
> **Response to review**
>
> We thank the reviewer for the detailed and very useful feedback. We have addressed all of your concerns below.
>
> Issue 0
> “..there are only 30 participants per condition and so it’s hard to tell whether the large differences in conditions are due to noise and what a stable ranking of conditions actually looks like …”
> A: Good point! As per your suggestion, we have increased the sample size substantially by recruiting a total of 120 subjects per condition. The results and conclusions remain unchanged.
>
> “... the error bars in figure 1 represent, are they standard deviations of the mean?... ”
> A: Sorry for the confusion. The error bars in Figure 1 represent standard error of the mean (we have added that clarification in the revision).
>
> “Did you collect any extra data about participants? One potentially helpful example is asking how familiar participants are with platformer video games …”
> A: Yes! For all of the games, we found only a moderate correlation (around 0.3) between familiarity with video games and average time taken to solve the game. This relatively moderate correlation indicates that familiarity with video games only results in slight improvement in performance of human players.
>
> Issue 1
> “What do you mean by “objects”?”
> A: Thank you for asking this question. We have clarified the definition of objects in the revised version of the manuscript. In the video game setting, objects are simply entities that are visibly distinct from their surroundings. The hypothesis is that humans use these visually distinct entities as subgoals, which results in more efficient exploration than random search. Performance of humans in game manipulation shown in Figure 2(c), demonstrates that when players cannot distinguish entities from the background, their performance drops significantly. We believe that mechanisms to bias exploration towards salient entities would be an interesting step towards improving the efficiency of RL agents.
>
> Issue 2.
> “There are two ways to interpret the authors’ main claim: the strong version would maintain that semantic priors aren’t important at all..”
> A: We are sorry for the confusion. Our claim is that while prior knowledge about semantics and affordances is important for human players, more general priors about objects (i.e. existence of visually salient entities that are subgoals; entities that look similar have the same semantics)  are more critical to performance. In essence, we agree with the reviewer and we do not claim that semantic priors aren't important (just that general prior about objects are more critical). We have revised the manuscript to clarify this. As per your suggestion, we have also included an additional experiment (refer to section A in Appendix) and indeed find that reversing the semantics leads to worse performance than that of simply masking the semantics.
>
> “..Here without semantic priors I would hypothesize that human performance would fall quite far (whereas with semantics people would be able to figure it out quite well).”
> A: We completely agree. However, at the same time, we believe that the prior of treating visually distinct entities as sub-goals for exploration will be more important than the prior about semantics alone.
>
> “..It’s also important to take into account how much work general priors about video game playing (games have goals, up jumps, there is basic physics) are doing here..”
> A: Great point. Humans bring in various priors about general video game playing such as moving up or right in games is generally correlated with progress, games have goals etc. Quantifying the importance of such priors is an interesting direction of research and we will include this discussion in the next revision of the paper.

---

> > ### Author Response · Authors · 2018-01-23
> > **Request for feedback**
> >
> > We have revised our paper and added new experiments to address all of your previous concerns. It would be great if you can please find some time to look at our response and inform us of any other feedback or concerns. This would go a long way in helping us improve the paper further. Thank you so much once again!

---

### Official Review · AnonReviewer2 · 2017-11-22
**I have concerns about the method and the conclusions**

**Rating:** 4
**Confidence:** 3

**Review:**

This paper investigates human priors for playing video games.

Considering a simple video game, where an agent receives a reward when she completes a game board, this paper starts by stating that:
-	Firstly, the humans perform better than an RL agent to complete the game board.
-	Secondly, with a simple modification of textures the performances of human players collapse, while those of a RL agent stay the same.

If I have no doubts about these results, I have a concern about the method.
In the case of human players the time needed to complete the game is plotted, and in the case of a RL agent the number of steps needed to complete the game is plotted (fig 1). Formally, we cannot conclude that one minute is lesser than 4 million of steps.

This issue could be easily fixed. Unfortunately, I have other concerns about the method and the conclusions.

For instance, masking where objects are or suppressing visual similarity between similar objects should also deteriorate the performance of a RL agent. So it cannot be concluded that the change of performances is due to human priors. In these cases, I think that the change of performances is due to the increased difficulty of the game.

The authors have to include RL agent in all their experiments to be able to dissociate what is due to human priors and what is due to the noise introduced in the game.

---

> ### Author Response · Authors · 2018-01-05
> **Reviewer might have misunderstood our experimental setup and methodology**
>
>
> The reviewer says “..Formally, we cannot conclude that one minute is lesser than 4 million of steps..”
>
> A: In Figure 1 of the revised manuscript,  we have now reported the number of steps taken by both the RL agents and human players for direct comparison. Human players take three orders of magnitude fewer steps to solve the game. Further, please note that the main point of our work was not to compare absolute performance of humans against RL agents, but to show that the performance of human players changes significantly with re-rendering of the game which makes it hard for humans to use their prior knowledge, whereas the performance of RL agent is almost unchanged.
>
> The reviewer says “..So it cannot be concluded that the change of performances is due to human priors. In these cases, I think that the change of performances is due to the increased difficulty of the game.”
>
> A: We are afraid that the reviewer might have misunderstood our experimental setup and methodology.  First, note that all games are *exactly the same* in their reward and goal structure - the only difference between the different versions of the game is in the rendering of the game entities. Because there is no other difference between the original and the manipulated versions of the game, it can be inferred that drop in performance is due to the inability of humans to employ their prior knowledge and beliefs in those manipulated games.
>
> The reviewer says, “...The authors have to include RL agent in all their experiments to be able to dissociate what is due to human priors and what is due to the noise introduced in the game”
>
> A: We do not agree with the reviewer’s comment because the performance of the RL agents on different manipulations of the game has no effect on the conclusions of the human study. At the same time, we do believe that studying the performance of RL agents on all game manipulations is an interesting question; one that is independent of the study of priors employed by humans. We have included the performance of RL agents for various game manipulations in Section C, Appendix. The RL agent’s performance is unaffected in all game manipulations except for the version in which visual similarity is removed. The results provide direct evidence that reviewer’s claim that “masking where objects are ...should also deteriorate the performance of a RL agent.” is simply not true.

---

> > ### Author Response · Authors · 2018-01-19
> > **Response to updated review rating**
> >
> > Thank you for upgrading the rating of our paper. We addressed the concerns raised in your review and reported all experiments you asked for.  It would be great and helpful if you could provide details about your current concerns. Thanks a lot for your time!

---

### Official Review · AnonReviewer3 · 2017-11-30
**Review - Accept**

**Rating:** 7
**Confidence:** 4

**Review:**

The authors present a study of priors employed by humans in playing video
games -- with a view to providing some direction for RL agents to be more
human-like in their behaviour.

They conduct a series of experiments that systematically elides visual
cues that humans can use in order to reason about actions and goals in a
platformer game that they have a high degree of control over.

The results of the experiments, conducted using AMT participants, demonstrates
the existence of a taxonomy of features that affect the ability to complete
tasks in the game to varying degrees.

The paper is clearly written, and the experiments follow a clean and coherent
narrative. Both the premises assumed and the conclusions drawn are quite
reasonable given the experimental paradigm and domain in which they are
conducted.

There were a couple of concerns I did have however:

1. Right at the beginning, and through the manuscript, there is something of an
   apples-to-oranges comparison when considering how quickly humans can
   complete the task (order of minutes) and how quickly the SOTA RL agents can
   complete the task (number of frames).

   While the general spirit of the argument is somewhat understandable despite
   this, it would help strengthen any inference drawn from human   performance
   to be applied to RL agents, if the comparison between the two were to be
   made more rigorous -- say by estimating a rough bijection between human and
   RL measures.

2. And in a related note to the idea of establishing a comparison, it would be
   further instructive if the RL agents were also run on the different game
   manipulations to see what (if any) sense could be made out of their
   performance.

   I understand that at least one such experiment is shown in Figure 1 which
   involves consistent semantics, but it would be quite interesting to see how
   RL agents perform when this consistency is taken away.

Other questions and comments:

1. In the graphs shown in Figure 3, are the meaning of the 'State' variable is
   not clear -- is it the number of *unique* states visited? If not, is it the
   total number of states/frames seen? In that case, how is it different from
   'Time'?

2. The text immediately below Figure 3's caption seems to have an incorrect
   reference (referring to Figure 2(a) instead of Figure 3(a)).

Given recent advances in RL and ML that eschew all manner of structured
representations, I believe this is a well-timed reminder that being able to
transfer know-how from human behaviour to artificially-intelligent ones.

---

> ### Author Response · Authors · 2018-01-05
> **Response to review**
>
> We thank the reviewer for the positive and useful feedback. Our response to your concerns below:
>
> Q: “there is something of an apples-to-oranges comparison when considering how quickly humans can complete the task (order of minutes) and how quickly the SOTA RL agents can complete the task (number of frames).”
> A:  Good point! In Figure 1 of the revised manuscript, we have now reported number of actions taken by both human players and RL agents to solve the games.
>
> Q: “... it would be further instructive if the RL agents were also run on the different game manipulations…”
> A: Thank you for this useful suggestion! We have included additional experiments that quantify the performance of  RL agent on the different game manipulations (Section C, Appendix). The RL agent’s performance is unaffected in all game manipulations except for the version in which visual similarity is removed.
>
> Response to additional comments:
> Q: “ .. graphs shown in Figure 3, are the meaning of the 'State' variable is not clear -- is it the number of *unique* states visited? ..”
> A: In graph 3, the 'State' variable indeed refers to the unique states visited which serves as a measure of how much players explore a game manipulation. We have clarified this in the revised manuscript.
>
> We have made revisions to the text addressing other minor corrections pointed by you.

---

### Author Response · Authors · 2018-01-05
**Main Response to Reviewers**

We thank the reviewers for their encouraging comments. We are glad that the reviewers found the questions addressed in the paper to be super important (R3) and the narrative to be coherent (R1). R1 says, “Given recent advances in RL ... it is a well-timed reminder that being able to transfer know-how from human behaviour to artificially-intelligent ones”.  The reviewers also had a number of great suggestions that we have incorporated in the revised manuscript.  The major changes are as follows:

a) We have rewritten the introduction to clarify the main claims of our paper.

b) We increased the sample size significantly for all the human experiments to ensure robustness.

c) We evaluated the performance of RL agent on various game manipulations to shed further light as to how RL agents differ from humans in terms of prior knowledge (Section C, Appendix).

---

### Decision · Program_Chairs · 2018-01-29
**ICLR 2018 Conference Acceptance Decision**

**Decision:**

Invite to Workshop Track

**Comment:**

This paper turned out to be quite difficult to call.  My take on the pros/cons is:

1. The research topic, how and why humans can massively outperform DQN, is unanimously viewed as highly interesting by all participants.

2. The authors present an original human subject study, aiming to reveal whether human outperformance is due to human knowledge priors.  The study is well conceived and well executed.  I consider the study to be a contribution by itself.

3. The study provides prima facie evidence that human priors play a role in human performance, by changing the visual display so that the priors cannot be used.

4. However, the study is not definitive, as astutely argued by AnonReviewer2.  Experiments using RL agents (with presumably no human priors) yield behavior that is similar to human behavior.  So it is possible that some factor other than human prior may account for the behavior seen in the human experiments.

5. It would indeed be better, as argued by AnonReviewer2, to use some information-theoretic measure to distinguish the normal game from the modified games.

6. The paper has been substantially improved and cleaned up from the original version.

7. AnonReviewer1 provided some thoughtful detailed discussion of how the authors may be overstating the conclusions that one can draw from the paper.

Bottom line: Given the procs and cons of the paper, the committee recommends this for workshop.